# Functional and structural characterization of an ECF-type ABC transporter for vitamin B12

**Joana A Santos**[1†], **Stephan Rempel**[1†], **Sandra TM Mous**[1], **Cristiane T Pereira**[2], **Josy ter Beek**[1‡], **Jan-Willem de Gier**[3], **Albert Guskov**[1*], **Dirk J Slotboom**[1,4*]

[1]Groningen Biomolecular and Biotechnology Institute (GBB), University of Groningen, Groningen, The Netherlands; [2]Institute of Biology, University of Campinas, Campinas, South America; [3]Department of Biochemistry and Biophysics, Center for Biomembrane Research, Stockholm University, Stockholm, Sweden; [4]Zernike Institute for Advanced Materials, University of Groningen, Groningen, The Netherlands

**\*For correspondence:**
a.guskov@rug.nl (AG);
d.j.slotboom@rug.nl (DJS)

[†]These authors contributed equally to this work

**Present address:** [‡]Umeå universitet, Umeå, Sweden

**Competing interests:** The authors declare that no competing interests exist.

**Abstract** Vitamin B12 (cobalamin) is the most complex B-type vitamin and is synthetized exclusively in a limited number of prokaryotes. Its biologically active variants contain rare organometallic bonds, which are used by enzymes in a variety of central metabolic pathways such as L-methionine synthesis and ribonucleotide reduction. Although its biosynthesis and role as co-factor are well understood, knowledge about uptake of cobalamin by prokaryotic auxotrophs is scarce. Here, we characterize a cobalamin-specific ECF-type ABC transporter from *Lactobacillus delbrueckii*, ECF-CbrT, and demonstrate that it mediates the specific, ATP-dependent uptake of cobalamin. We solved the crystal structure of ECF-CbrT in an *apo* conformation to 3.4 Å resolution. Comparison with the ECF transporter for folate (ECF-FolT2) from the same organism, reveals how the identical ECF module adjusts to interact with the different substrate binding proteins FolT2 and CbrT. ECF-CbrT is unrelated to the well-characterized B12 transporter BtuCDF, but their biochemical features indicate functional convergence.
DOI: https://doi.org/10.7554/eLife.35828.001

## Introduction

Vitamin B12 or cobalamin (Cbl) is regarded as the largest and most complex biological 'small mole-cule'. The molecule consists of a corrin ring chelating a cobalt ion using four equatorial coordinating nitrogen atoms (*Figure 1*, *Figure 1—figure supplement 1*). At the α-axial position, the cobalt ion is coordinated by the 5,6-dimethylbenzimidazole (DMBI) base that is linked covalently to the corrin ring. Located at the β-axial position is the sixth coordinating moiety (*Gruber et al., 2011*; *Roth et al., 1996*). This ligand can vary among cobalamin derivatives and forms a rare organometal-lic, covalent bond, which offers unique catalytic properties to enzymes that use Cbl as a co-factor. The two most common biological active variants have a methyl or 5'-deoxyadenosyl group at this position, resulting in methyl- and adenosyl-cobalamin (Met-Cbl and Ado-Cbl, respectively). In the industrially produced variant, a cyano-group (CN-Cbl) binds at the β-axial position, and a hydroxy group (OH-Cbl) is present in the degradation product (*Gruber et al., 2011*).

Enzymes that use Cbl as their co-factor catalyze mostly methyl group transfer reactions, or a vari-ety of different radical-mediated reactions (*Giedyk et al., 2015*; *Gruber et al., 2011*). The most prominent example for methyl group transfer is MetH, the Cbl-dependent L-methionine synthase, which uses Met-Cbl to transfer a methyl group onto L-homocysteine to produce L-methionine. The

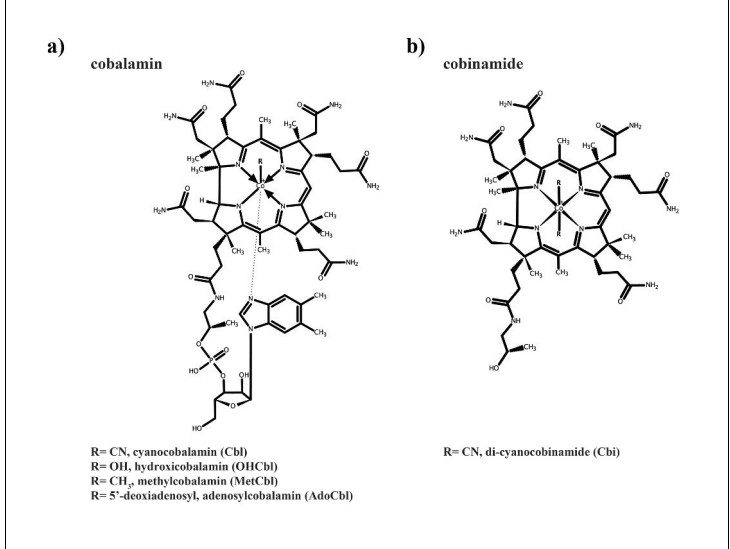

**Figure 1.** Structures of cobalamin and cobinamide. (**a**) Cobalamin structure, represented in the base-on conformation with the 5',6'-dimethyl-benzimidazole ribonucleotide moiety (α-ligand) coordinating the central cobalt ion. The variable β-ligands are denoted as R in the lower left corner. (**b**) Structure of cobinamide, which lacks the DMBI moiety and has two cyano groups coordinating the cobalt ion from each side of the corrin ring.
DOI: https://doi.org/10.7554/eLife.35828.002

The following figure supplement is available for figure 1:

**Figure supplement 1.** 3D structures of cobalamin and cobinamide.
DOI: https://doi.org/10.7554/eLife.35828.003

methyl group on Cbl is subsequently restored from methyl-folate (*Banerjee and Matthews, 1990*; *Gruber et al., 2011*).

The ability to synthesize Cbl de novo is restricted to prokaryotic species in only ~20 genera. Two routes for de novo synthesis have been established (aerobic or anaerobic), each requiring approximately 30 different enzymes, which makes de novo synthesis a very energy consuming process, and could explain why roughly two thirds of prokaryotes that require Cbl cannot synthesize the molecule, and hence depend on uptake (*Gruber et al., 2011*; *Roth et al., 1996*). For some microbial communities, for instance in the Ross Sea, it has been shown that Cbl production is the limiting factor for biomass production, which generates a demand for Cbl uptake systems among Cbl auxotrophs (*Bertrand et al., 2011*).

In contrast to the well-characterized chemical properties of Cbl, as well as its biosynthesis and role in many enzymatic reactions, the uptake of the vitamin by bacteria is poorly understood. The only characterized Cbl uptake system is the *Escherichia coli* BtuCDF ATP binding cassette (ABC) transporter, which was first described in 1980 (*DeVeaux and Kadner, 1985*). Substantial understanding of the system has been obtained through a combination of biochemical and structural studies (*Borths et al., 2002*; *Goudsmits et al., 2017*; *Korkhov et al., 2012*; *Korkhov et al., 2014*; *Locher et al., 2002*). The importer uses the periplasmic substrate-binding protein BtuF to capture Cbl or its precursor cobinamide (Cbi) with high affinity ($K_d$ values of ~10 nM and ~40 nM, respectively) (*Cadieux et al., 2002*; *Mireku et al., 2017*). Transport is powered by hydrolysis of ATP by the two BtuD subunits located on the cytoplasmic side of the membrane (*Borths et al., 2005*). The substrate passes through the membrane at the interface between two copies of the transmembrane protein, BtuC (*Korkhov et al., 2012*; *Korkhov et al., 2014*). BtuCDF homologs are found widely in prokaryotes, but they are absent from a subset of bacteria that require uptake of vitamin B12 (*Rodionov et al., 2009*).

An in silico study by *Rodionov et al. (2009)* predicted that the energy coupling factor (ECF-) type ABC transporter ECF-CbrT might be a Cbl transporter (*Rodionov et al., 2009*). ECF-transporters are multi-subunit membrane complexes that consist of two ATPases, similar to the ATPases of ABC transporters, and two membrane embedded proteins, not related to any other protein family

(*Slotboom, 2014*). The two ATPases and one of the transmembrane proteins, EcfT, form the 'energizing unit' or ' ECF-module'. The other membrane protein, termed S-component, acts as the substrate-binding protein and dynamically associates with the ECF-module to allow for substrate translocation. In so-called group II ECF transporters, multiple S-components specific for different substrates interact with the same ECF module (*Berntsson et al., 2012*; *Henderson et al., 1979*; *Karpowich et al., 2015*; *Majsnerowska et al., 2015*; *ter Beek et al., 2011*). For instance, in *Lactobacillus delbrueckii* eight different S-components are predicted to share a single ECF module, one of which, CbrT, was predicted to be specific for Cbl (*Rodionov et al., 2009*; *Swier et al., 2016*).

In this work, we biochemically and structurally characterize the ECF-CbrT complex from *L. delbrueckii*. We demonstrate that ECF-CbrT is a Cbl transporter that catalyses ATP-dependent uptake of Cbl and its precursor Cbi. We show that the S-component CbrT mediates high affinity substrate-specificity for Cbl and Cbi, and we report the crystal structure of ECF-CbrT from *L. delbrueckii* at 3.4 Å resolution in its *apo* inward-facing state. Although ECF-CbrT is structurally and mechanistically unrelated to BtuCDF, the kinetic parameters of the two transporters are very similar, suggestive of functional convergence.

## Results

### Expression of ECF-CbrT complements an *Escherichia coli* strain lacking its endogenous vitamin B12 transporter

To demonstrate that ECF-CbrT is a vitamin B12 transporter, we constructed an *Escherichia coli* knock-out strain with three genomic deletions: Δ*btuF*, Δ*metE*, and Δ*btuC::Km^R* (*E. coli* ΔFEC) (*Baba et al., 2006*; *Datsenko and Wanner, 2000*; *Thomason et al., 2007*). A similar strain was previously used by *Cadieux et al., 2002* to identify the substrate binding protein BtuF. The knock-out strain lacks the L-methionine synthase MetE (*Davis and Mingioli, 1950*). *E. coli* possesses two L-methionine synthases, MetE and MetH. MetH uses Cbl as cofactor, whereas MetE is not dependent on the vitamin (*Banerjee et al., 1989*; *Davis and Mingioli, 1950*). Thus, deletion of *metE* makes *E. coli* dependent on Cbl for the synthesis of L-methionine. Because *btuF* and *btuC* are also deleted in *E. coli* ΔFEC, endogenous Cbl-uptake mediated by BtuCDF is abolished (*Cadieux et al., 2002*), and (heterologous) expression of an active Cbl transporter is required to synthesize methionine.

We studied the growth of the deletion strain transformed with an expression plasmid for either BtuCDF (positive control), or an empty plasmid (negative control), or CbrT with or without the ECF module. We grew cells in 96-well plates using minimal medium supplemented with L-methionine or Cbl and monitored the optical density at 600 nm (OD$_{600}$). In the presence of L-methionine, all strains grew readily, with a lag phase of 300–450 min (*Figure 2a–d*) (*Zwietering et al., 1990*). The strains expressing BtuCDF or ECF-CbrT showed similar growth characteristics (lag time of 380 min and 470 min, respectively) in medium containing Cbl instead of L-methionine (*Figure 2a and c*, respectively), whereas the deletion strain expressing solitary CbrT (without its cognate ECF-module) did not show substantial growth in the absence of L-methionine when supplemented with Cbl (*Figure 2d*). The results demonstrate that the full ECF-CbrT complex constitutes a new Cbl transporter. ECF-CbrT also supported growth of *E. coli* ΔFEC in the presence of Cbi instead of Cbl, albeit with a longer lag-time (730 min) (*Figure 2c*), indicating that Cbi is a transported substrate of ECF-CbrT. We hypothesize that the longer lag time is due to the extra time required to express the necessary enzymes for Cbl synthesis from Cbi (*Lawrence and Roth, 1995*). Our *in vitro* work shows that ECF- CbrT indeed supports efficient Cbl and Cbi transport (see below).

### ECF-CbrT catalyzes ATP-dependent transport of cobalamin and cobinamide

We purified ECF-CbrT, reconstituted the complex in liposomes and assayed for the uptake of radio-labeled Cbl ($^{57}$Co-cyanocobalamin). Uptake of radiolabeled Cbl into proteoliposomes was observed only when the proteoliposomes were loaded with Mg-ATP and not when Mg-ADP was incorporated (*Figure 3a*). While this experiment shows that transport is strictly ATP-dependent, similar to what was found for other ECF transporters (*Swier et al., 2016*; *ter Beek et al., 2011*), the ratio between ATP molecules hydrolyzed and Cbl molecules transported cannot be derived from this data. To

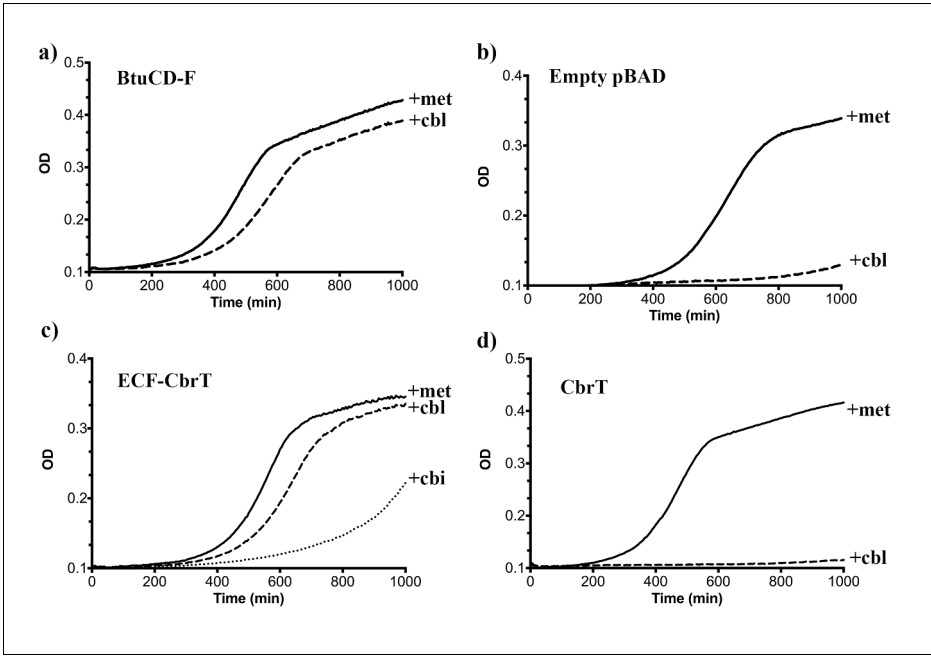

**Figure 2.** ECF-CbrT supports cobalamin-dependent growth of an *E.coli* deletion strain. (a) The triple knock out strain *E. coli* ΔFEC expressing the BtuCDF ABC-transporter (positive control) grows in the presence of 50 μg/ml L-methionine or 1 nM CN-Cbl with lag-times of 300 min or 380 min, respectively, (b) *E. coli* ΔFEC carrying only the empty expression vector (negative control) grows only in the presence of 50 μg/ml L-methionine but not with 1 nM CN-Cbl. The lag-times of the negative controls are 450 min or >1000 min, respectively. (c) *E. coli* ΔFEC expressing the entire ECF-CbrT transporter supports growth in the presence of either 1 nM Cbl or 1 nM Cbi with a lag-time of 470 min or 730 min, respectively. The lag-time in the presence of 50 μg/ml L-methionine is 410 min. (d) Expression of the solitary S-component CbrT without its cognate ECF-module is not able to support growth of *E. coli* ΔFEC in the presence of 1 nM CN-Cbl.

DOI: https://doi.org/10.7554/eLife.35828.004

obtain this ratio, simultaneous measurements of Cbl uptake and ATP hydrolysis rates are needed, which is technically difficult. Additionally, the related ECF transporter for folate displays a large extent of futile ATP hydrolysis (not coupled to transport [*Swier et al., 2016*]), which further complicates the determination of the coupling ratio. Nonetheless, the EcfA and EcfA' subunits contain all the motifs to form functional ATPases, and therefore we speculate that transport of Cbl is coupled to the hydrolysis of two ATP molecules. Using an Mg-ATP concentration of 5 mM, the apparent $K_M$ for Cbl uptake was 2.1 ± 0.4 nM and the $V_{max}$ = 0.06 ± 0.01 nmol × mg$^{-1}$ × s$^{-1}$ (*Figure 3—figure supplement 1*). To test the substrate specificity of the new vitamin B12 transporter, we conducted uptake experiments with a variety of competing compounds that are structurally similar to Cbl (*Figure 3b*). Addition of a 250-fold excess of unlabeled CN-Cbl inhibited the uptake of the radiolabeled substrate almost completely. Met-Cbl and OH-Cbl inhibited uptake to a similar extent as CN-Cbl, whereas Ado-Cbl was less effective (inhibition to ~25%). Addition of a 250-fold excess of Cbi also decreased the uptake of radiolabeled Cbl to ~25% (*Figure 3b*). To test whether Cbi is a transported substrate (that competitively inhibits transport of Cbl) or a non-transported compound that can only bind to the transporter, we directly measured uptake of Cbi (*Figure 3a*). Radiolabeled Cbi is not commercially available and, therefore, we synthesized the compound by treating $^{57}$Co-cyano-cobalamin with perchloric acid (Schneider and Stroiński, 1987). The complete conversion of Cbl into Cbi was confirmed by mass spectrometry. Cbi was transported into liposomes containing ECF-CbrT, and transport required lumenal Mg-ATP, confirming that Cbi is a transported substrate (*Figure 3a*). Finally, we tested whether hemin inhibits Cbl transport (*Figure 3b*). Hemin and Cbl are structurally related and share the same precursor uroporphyrinogen-III (*Roth et al., 1996*), but unlike Cbl, hemin consists of a flat porphyrin ring with a chelated iron ion, and has a chloride ion as one of the axial

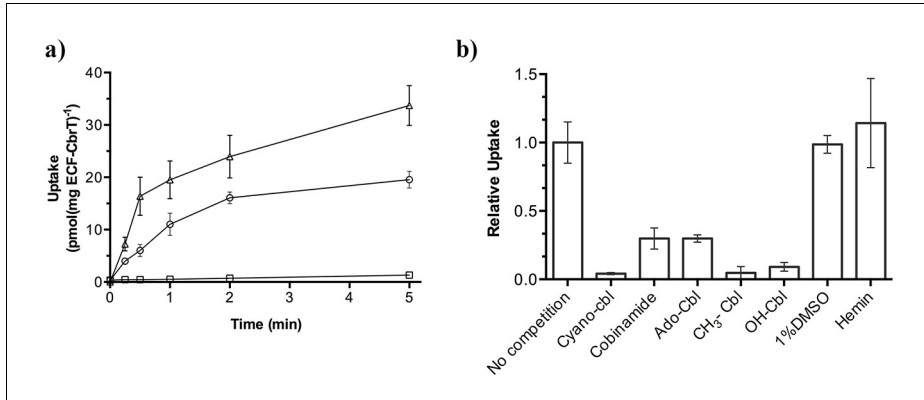

**Figure 3.** $^{57}$Co-cyanocobalamin (cyano-Cbl) and $^{57}$Co-cobinamide (Cbi) transport by purified and reconstituted ECF-CbrT. (a) ATP-dependent uptake of radiolabeled CN-Cbl and Cbi by ECF-CbrT in proteoliposomes. Proteoliposomes were loaded with either 5 mM Mg-ATP (circles for CN-Cbl, triangles for Cbi) or 5 mM Mg-ADP (CN-Cbl, squares). (b) Competition assay using Cbl-analogues. The initial uptake rate at 1 nM $^{57}$Co-cyanocobalamin (CN-Cbl) was measured. Competing compounds (adenosyl-cobalamin (Ado-Cbl); methyl-cobalamin (CH$_3$-Cbl); hydroxyl-cobalamin (OH-Cbl); cobinamide or hemin) were added at a concentration of 250 nM. The uptake was normalized to a condition without competitor (10 pmol*mg$^{-1}$*min$^{-1}$). Since hemin is not readily soluble in an aqueous solution, we added 1% (v/v) DMSO during the assay, which did not affect the transporter activity. All competition experiments were performed in triplicate and the error bars indicate the standard deviation (s.d.).

DOI: https://doi.org/10.7554/eLife.35828.005

The following figure supplement is available for figure 3:

**Figure supplement 1.** Kinetics of cobalamin uptake by ECF-CbrT.

DOI: https://doi.org/10.7554/eLife.35828.006

ligands. Hemin did not compete with Cbl-uptake (*Figure 3b*), showing that, although promiscuous among Cbl variants and Cbl-precursors, ECF-CbrT is a dedicated vitamin B12 transporter.

We aimed to obtain further biochemical information on the solitary S-component CbrT. We could purify CbrT only in the substrate-bound state (*Figure 4a*). The protein without substrate was unstable in detergent solution and prone to aggregation. Apparently, substrate binding had a stabilizing effect on CbrT, an observation that has been made more often for membrane proteins (*Berntsson et al., 2012*; *ter Beek et al., 2011*).

From the spectral properties of Cbl that was co-purified with CbrT, we conclude that CbrT binds Cbl in a ~ 1:1 ratio in detergent solution (*Figure 4a*), which reflects the common substrate to protein stoichiometry for S-components (*Berntsson et al., 2012*; *Erkens et al., 2011*; *Swier et al., 2016*). Because we could not obtain purified *apo* CbrT, we studied the substrate-binding affinity using *E. coli* crude membrane vesicles (CMVs) containing overexpressed *apo* CbrT. Isothermal titration calorimetry (ITC) revealed binding of both CN-Cbl and Cbi with dissociation constants of 9.2 ± 4.5 nM and 36 ± 15 nM, respectively (*Figure 4b,c and e*). As a negative control, CMVs without CbrT were used to exclude unspecific binding of CN-Cbl or Cbi (*Figure 4d*). Cbl analogues, OH-Cbl and Met-Cbl, were also probed with ITC and found to bind to CbrT with the similar binding affinities as CN-Cbl (K$_d$ values of 9.6 ± 6.9 nM or 4.5 ± 0.3 nM, respectively, *Figure 4—figure supplement 1*), supporting the notion that CbrT is promiscuous towards the β-axial ligand of Cbl and corroborating the findings of the competition assay.

Although the use of crude membrane vesicles precluded the determination of the number of binding sites (the concentration of CbrT in the membrane vesicles is unknown), the thermodynamic values (K$_d$, $\Delta H$ and $\Delta S$) derived from the ITC measurements do not depend on this number. Assuming that CbrT has a single substrate binding site (consistent with the spectral properties, *Figure 4a*), the expression level of CbrT in the membranes could be calculated and we found that CbrT accounted for ~0.9% (w/w) of the protein content in the membrane.

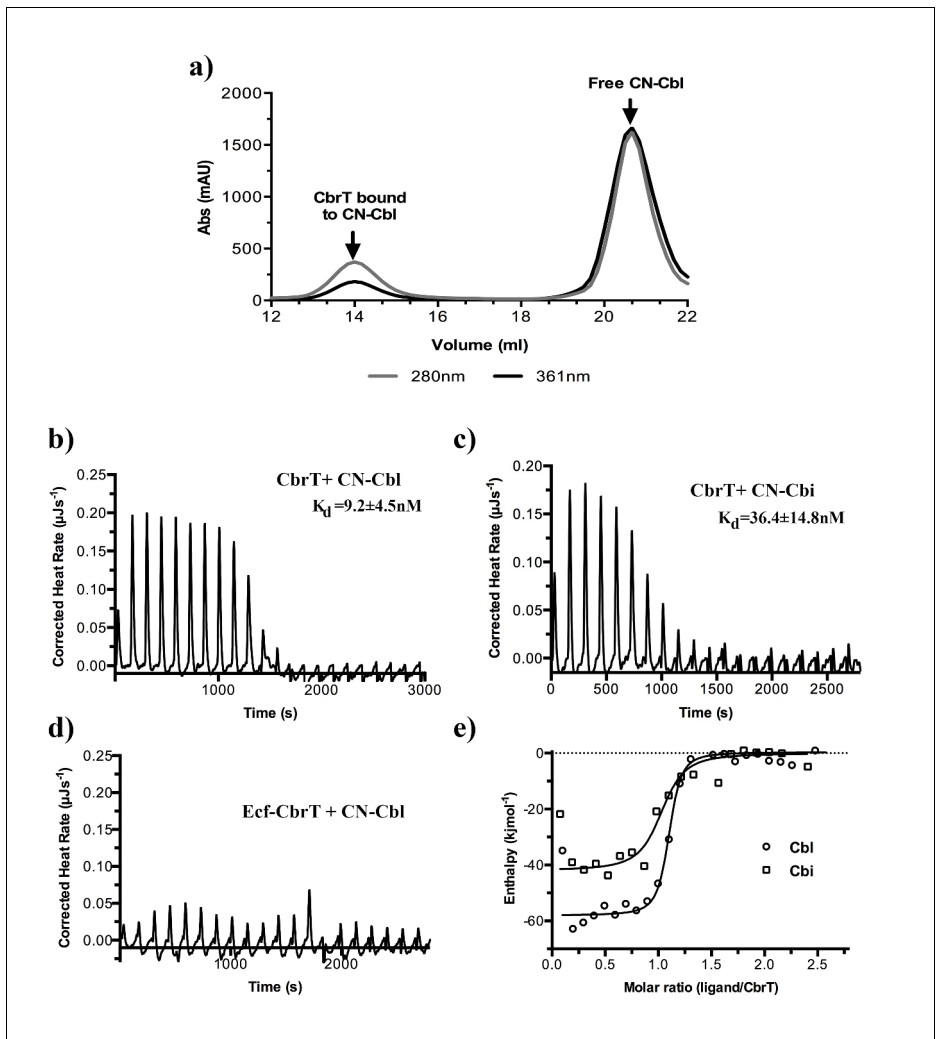

**Figure 4.** Cobalamin and cobinamide binding to CbrT. (**a**) Co-purification of CN-cobalamin with CbrT. The elution peak of the size exclusion column at a volume of 14 ml contains purified CbrT. The protein absorbs at 280 nm and CN-Cbl at 361 nm, showing that CbrT is eluted bound to CN-Cbl. (**b**) and (**c**) ITC measurements of Cbl and Cbi binding to CbrT. The determined $K_d$ values for Cbl and Cbi were averaged from triplicate measurements and the error is s.d. (**d**) ITC measurement showing the absence of Cbl-binding to the full complex, ECF-CbrT. Fitting of single binding site models to the data is shown in panel (**e**).

DOI: https://doi.org/10.7554/eLife.35828.007

The following figure supplement is available for figure 4:

**Figure supplement 1.** Binding of Cbl-analogs to CbrT.

DOI: https://doi.org/10.7554/eLife.35828.008

## Structure of the vitamin B12-specific ECF transporter in its *apo* and post substrate-release state

We crystallized the ECF-CbrT complex in detergent (n-Dodecyl-β-D-maltopyranoside, DDM) solution and solved a crystal structure to 3.4 Å resolution using molecular replacement with the structure of the folate transporter, ECF-FolT2, from *L. delbrueckii* as a search model (*Figure 5a*) (*Swier et al., 2016*). Statistics of data collection and structure refinement are summarized in *Supplementary file 1*. There are two copies of ECF-CbrT in the asymmetric unit, corresponding to molecules A and B, each of them comprising CbrT, EcfT, EcfA and EcfA'.

The identical ECF modules of the ECF-FolT2 and ECF-CbrT complexes have very similar overall structures, with a few notable conformational differences (*Figure 5b*). In both complexes, the two

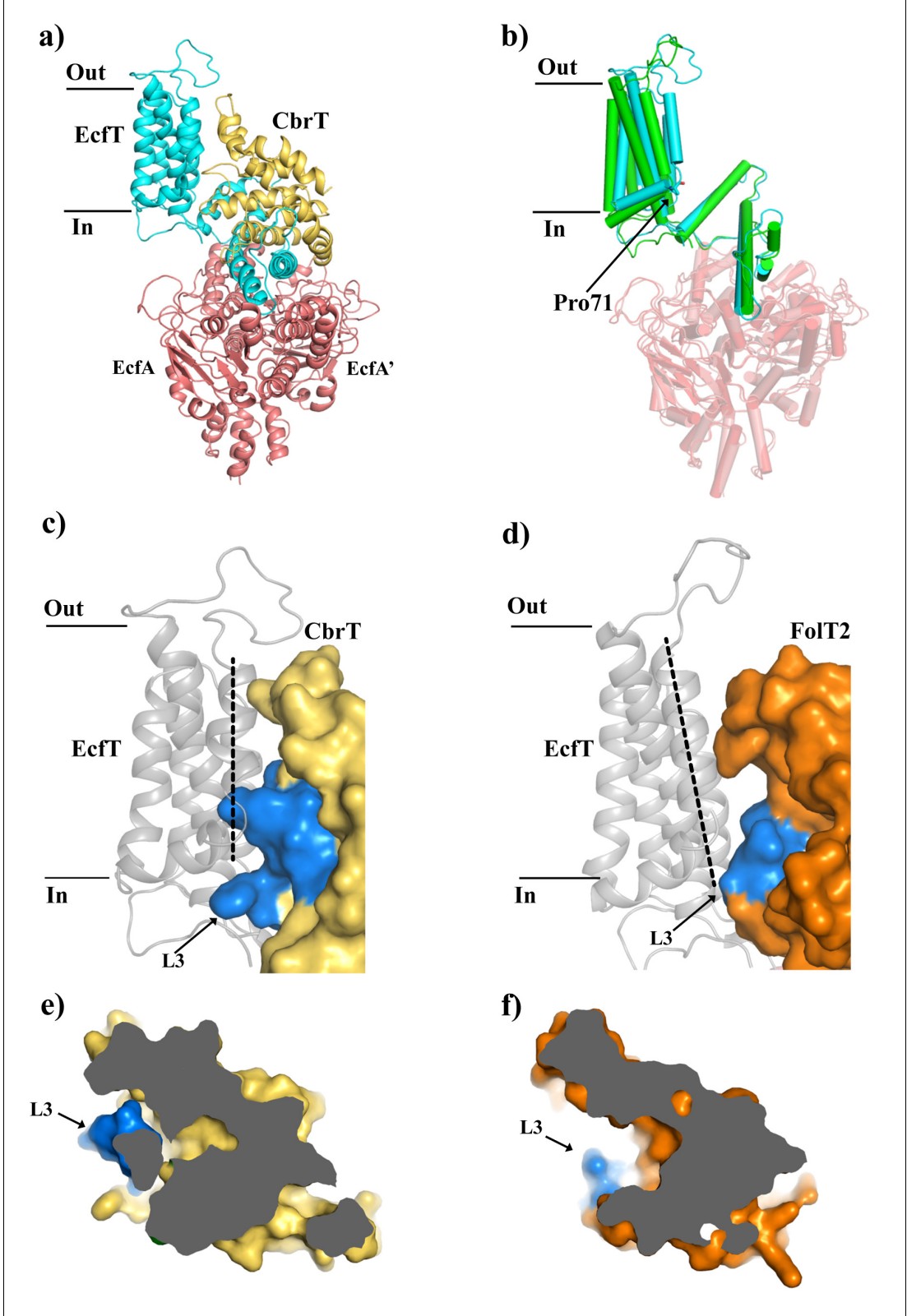

**Figure 5.** Comparison of the structures of ECF-CbrT and ECF-FolT from *L.delbrueckii*. (a) Cartoon representation of ECF–CbrT from the perspective of the plane of the membrane. Cytoplasmic ATPases, EcfA and EcfA', are colored in red, EcfT in cyan and CbrT in yellow. (b) Structural differences between the membrane domains of EcfT. The structures of ECF-CbrT (cyan) and ECF-FolT2 (green) from *L. delbrueckii* were superimposed by structural alignment of the ATPase units. Pro71 of EcfT is represented in sticks. (c) and (d) Surface representation of CbrT (c, yellow) and FolT2 (d, orange)
*Figure 5 continued on next page*

*Figure 5 continued*

interacting with EcfT (cartoon representation coloured in grey) with loop 3 of the S-components colored in blue. A dashed line highlights the movement of transmembrane helix 3 of EcfT. (**e**) and (**f**) Loop 3 obstructs access to the substrate binding cavity in CbrT but not in FolT2. (**e**) Slice-through of CbrT in surface representation, viewed from the plane of the membrane. Loop 3 is colored in blue. The ECF module has been omitted for clarity. (**f**) Same slice through representation like in (**e**) but for FolT2.

DOI: https://doi.org/10.7554/eLife.35828.009

nucleotide-binding domains (NBDs; EcfA and EcfA') are in a nucleotide-free state, adopting an open conformation with two incomplete ATP-binding sites. With the NBDs of ECF-FolT2 and ECF-CbrT aligned structurally (rmsd 1.8 Å), the coupling helices of EcfT, which transmit the conformational changes of the NBDs upon ATP-hydrolysis to the membrane domains, also superimpose well between the two complexes (rmsd 1.5 Å). However, the transmembrane-helices of EcfT adopt different conformations (rmsd 4.4 Å). They are offset like rigid bodies, hinging approximately around Pro71 (*Figure 5b*). Structural flexibility of the membrane domain has been observed before (*Swier et al., 2016*; *Zhang et al., 2014*) and is likely necessary to accommodate different S-components, and may facilitate toppling of the S-components during the catalytic cycle (*Swier et al., 2016*).

The S-components CbrT and FolT2, which interact with the same ECF module in *L. delbrueckii*, do not share significant sequence similarity (16% identical residues). Accordingly, the structures show pronounced differences (rmsd 3.1 Å), although the overall folds are the same. Particularly, differences in loop 3 and loop 5 cause alterations of the protein surfaces that interact with the membrane domain of EcfT (*Figure 5c,d*). Therefore, tight association of the different S-components with the same ECF module requires the conformational adaptations in the membrane domain of the interacting EcfT subunits (*Figure 5c,d*).

CbrT is in a 'toppled' orientation in the ECF-CbrT complex with TMs 1, 2, 3 and 4 oriented almost parallel to the membrane plane. Although OH-Cbl was added in excess to the crystallization condition, the substrate was not bound. The *apo*-state of the toppled S-component was observed before in other ECF-type transporters and likely represents the inward-facing state after substrate release (*Swier et al., 2016*; *Wang et al., 2013*; *Xu et al., 2013*; *Zhang et al., 2014*). The absence of substrate is in agreement with the proposed transport model, in that the inward oriented *apo* protein is a low affinity state (*Swier et al., 2016*). It has been hypothesized that this state precedes ATP hydrolysis, which leads to release of the S-component from the ECF module and reorientation of the S-component in the membrane, which brings the substrate binding site back to the extracellular side (*Swier et al., 2016*).

We hypothesize that the binding site for Cbl is located in a large cavity observed in the CbrT structure. The location of the cavity matches the position of the substrate-binding sites in structurally characterized S-components in the substrate-bound state (*Berntsson et al., 2012*; *Erkens et al., 2011*; *Swier et al., 2016*). In contrast to ECF-FolT2 (*Swier et al., 2016*), and structurally characterized ECF complexes from other organisms (*Wang et al., 2013*; *Xu et al., 2013*; *Zhang et al., 2014*), the binding cavity in ECF-CbrT is largely occluded, and not accessible from the cytosol (*Figure 5e,f*). The occlusion is caused mainly by the position of loop 3, which obstructs access to the cavity in CbrT. We speculate that occlusion of the empty binding site after cytoplasmic release of the substrate may be required for the subsequent reorientation of the S-component, upon release from the ECF module.

## Discussion

Comparative genomics studies have identified a wide range of ECF transporter families along with their putative substrate specificities mediated by their S-components (*Rodionov et al., 2009*). Based on its genetic organization and the lack of a BtuCDF transporter homolog in the *Lactobacillales* genomes, CbrT was predicted to be a vitamin B12-specific S-component (*Rodionov et al., 2009*). In *L. delbrueckii*, which was shown to be auxotrophic for vitamin B12 (*Kusaka and Kitahara, 1962*), CbrT occurs within the *nrdJ–cbrS–cbrT–pduO* gene cluster. This genetic organization strongly implies an involvement of the substrate-binding protein CbrT in Cbl uptake: First, *nrdJ* is annotated as an adenosylcobalamin-dependent ribonucleotide reductase requiring the vitamin as a co-factor.

Second, PduO is a Cbl adenosyltransferase, which converts Cbl into Ado-Cbl making it accessible for NrdJ. We showed that ECF-CbrT not only transports Cbl, but also mediates uptake of Cbi (*Figure 3a*). Therefore *L. delbrueckii* is expected to have the genetic repertoire to synthesize Cbl using Cbi as a precursor. However, we could not find homologs of the enzymes CobS, CobU and CobT, which are known to convert Cbi into cobalamin in *E. coli* (*Lawrence and Roth, 1995*). Possibly other proteins might functionally compensate for their absence, which was shown for the thiamin kinase YcfN, which could replace CobU in *Salmonella typhimurium* (*Otte et al., 2007*). However, it is also possible that *L. delbrueckii* CbrT binds Cbi in a futile mannerin which case Cbi would be transported but not used as a substrate for any enzyme. Finally, predicted CbrT homologs in *Lactobacillales* display a high degree of sequence identity, ranging from 25% to 60% compared to *L. delbrueckii* CbrT. Thus, we hypothesize that these CbrT homologs share the same function and represent substrate binding proteins for both Cbl and Cbi.

Our results show that ECF-CbrT is a new vitamin B12 transporter that is able to restore Cbl- and Cbi-dependent growth in *E. coli* ΔFEC (*Figure 2c*). Further characterization using uptake experiments with the purified ECF-CbrT complex (*Figure 3a*) and binding studies on CbrT show that the transporter is promiscuous towards the β-ligand of Cbl and also accepts Cbi as substrate (*Figure 3b*). A similar behavior has also been observed for BtuCDF (*Mireku et al., 2017*).

All three naturally occurring Cbl variants, OH-Cbl, Ado-Cbl and Met-Cbl, inhibit uptake of radiolabeled cyano-Cbl (*Figure 3b*). Whereas almost full inhibition was observed by a 250-fold excess of OH-Cbl, CN-Cbl, and Met-Cbl, Ado-Cbl inhibited only to 25%. This might be due to its bulkier β-ligand and a consequent steric hindrance. Nonetheless, the assay shows that ECF-CbrT is promiscuous toward the β-axial ligand of vitamin B12. Generally, poly-specificity to Cbl (and Cbl-derivatives, see below) and Cbi seems to be an inherent feature of vitamin B12 binding proteins, which is the case for BtuF, human Cbl-carriers, and also CbrT (*Fedosov et al., 1995*; *Mireku et al., 2017*). In the human Cbl-carriers that share a similar promiscuity, the β-ligand side of the bound substrate is partially solvent exposed (*Furger et al., 2013*; *Mathews et al., 2007*; *Wuerges et al., 2007*). In CbrT, the broader substrate specificity might be related to flexibility of loops 1 and 3 that are the gates of CbrT and would make contact with the varying β-axial ligands.

In other ECF-transporters, the S-components exhibit remarkable high affinities toward their respective substrates with $K_d$ values in the low nanomolar range (*Slotboom, 2014*). ITC measurements with CbrT in the absence of the ECF module also show high-affinity binding of CN-Cbl, OH-Cbl, Met-Cbl, and Cbi (*Figure 4b,c and e*). The slightly lower affinity for Cbi (4-fold) is probably due to the lack of the α-igand (*Figure 1—figure supplement 1*) that leads to fewer possible protein-substrate interactions.

Strikingly, the affinity binding constants for CN-Cbl and Cbi are in the same range as the respective affinities determined for BtuF (Cbl of 9.1 nM and Cbi of 40 nM) (*Mireku et al., 2017*), which might imply that Cbl-transporters evolved to acquire the substrates with similar efficiency. Together with our $V_{max}$ and $K_M$ (*Figure 3—figure supplement 1*) determination, we additionally show that the rate limiting step is substrate translocation, which means that the observed affinities are probably optimized for efficient substrate scavenging, followed by a slow translocation step. Human carriers achieve even higher affinities for Cbl (in the sub-picomolar range) (*Fedosov et al., 1995*), but in these cases, the *off* rate is practically zero and substrate release requires proteolysis, which is not the mechanism of (ABC-) transporters.

Although it was already known for a long time that a plethora of uncharacterized prokaryotic vitamin B12 uptake systems must exist, only the BtuCDF complex had been extensively characterized. This is somewhat surprising, considering the potential relevance of bacterial vitamin B12 transport for pharmaceutical applications. For instance, given the increase in antibiotic resistance and its serious threat to public health (*Alós, 2015*), it is imperative to find and characterize novel protein targets for drug design. Several pathogenic bacteria, such as *Streptoccoccus pyogenes* and *Clostridium tetani*, carry a *cbrT* gene, lack a BtuCDF homolog, and are Cbl-auxotrophs, which makes them strictly dependent on dedicated transporters to scavenge either vitamin B12 or its precursors from the environment. Because humans use endocytosis to take up Cbl (*Quadros, 2010*), Cbl-specific prokaryotic transporters are potential drug targets for vitamin B12 auxotrophic pathogens.

# Materials and methods

## Key resources table

| Reagent type (species) or resource | Designation | Source or reference | Identifiers | Additional information |
|---|---|---|---|---|
| Gene (*Lactobacillus delbrueckii* subsp. *bulgaricus*) | cbrT | NA | LDB_RS00385 | |
| Strain, strain background (*E. coli*) | MC1061 | Casadaban, M. J., and Cohen, S. N. (1980). Analysis of gene control signals by DNAfusion and cloning in Escherichia coli.Journal of Molecular Biology, 138(2),179–207 PMID 6997493 | | E. coli ΔFEC was constructed in this paper with the following deletions ΔbtuF, ΔmetE, and ΔbtuC::KmR. Strain requires either L-methionine or cobalmin/cobinamide plus expression of an appropiate cobalmin/cobinamide transporter. Strain can be made available upon reasonable request. |
| Strain, strain background (*E. coli*) | JW0154 | Coli Genetic Stock Center Yale | | E. coli ΔFEC was constructed in this paper with the following deletions ΔbtuF, ΔmetE, and ΔbtuC::KmR. Strain requires either L-methionine or cobalmin/cobinamide plus expression of an appropiate cobalmin/cobinamide transporter. Strain can be made available upon reasonable request. |
| Strain, strain background (*E. coli*) | JW3805 | Coli Genetic Stock Center Yale | | E. coli ΔFEC was constructed in this paper with the following deletions ΔbtuF, ΔmetE, and ΔbtuC::KmR. Strain requires either L-methionine or cobalmin/cobinamide plus expression of an appropiate cobalmin/cobinamide transporter. Strain can be made available upon reasonable request. |
| Strain, strain background (*E. coli*) | JW1701 | Coli Genetic Stock Center Yale | | E. coli ΔFEC was constructed in this paper with the following deletions ΔbtuF, ΔmetE, and ΔbtuC::KmR. Strain requires either L-methionine or cobalmin/cobinamide plus expression of an appropiate cobalmin/cobinamide transporter. Strain can be made available upon reasonable request. |
| Strain, strain background (*E. coli*) | ΔFEC | This paper | | E. coli ΔFEC was constructed in this paper with the following deletions ΔbtuF, ΔmetE, and ΔbtuC::KmR. Strain requires either L-methionine or cobalmin/cobinamide plus expression of an appropiate cobalmin/cobinamide transporter. Strain can be made available upon reasonable request. |
| Biological sample (*Lactobacillus delbrueckii*) | Lactobacillus delbrueckii subsp. bulgaricus genomic DNA | DSMZ | DSM 20081 | |
| Recombinant DNA reagent | pBAD24_CbrT | This paper | | Expression plasmids for CbrT and ECF-CbrT in *E. coli*. Plasmids can be provided upon reasonable request. |
| Recombinant DNA reagent | p2BAD_ECF_CbrT | This paper | | Expression plasmids for CbrT and ECF-CbrT in *E. coli*. Plasmids can be provided upon reasonable request. |
| Chemical compound, drug | CN-Cbl | Acros | 405920050 | |
| Chemical compound, drug | OH-Cbl | Sigma-Aldrich | 95200–1G | |
| Chemical compound, drug | Met-Cbl | Sigma-Aldrich | M9756-250G | |
| Chemical compound, drug | Ado-Cbl | Sigma-Aldrich | C0884-250MG | |

*Continued on next page*

*Continued*

| Reagent type (species) or resource | Designation | Source or reference | Identifiers | Additional information |
|---|---|---|---|---|
| Chemical compound, drug | Cbi | Sigma-Aldrich | C3021-50MG | |
| Chemical compound, drug | hemin | Sigma-Aldrich | 51280–1G | |
| Chemical compound, drug | 57Co-cyanocobalamin | MP-Biomedicals | 06B-430000 | |
| Chemical compound, drug | perchloric acid | Sigma-Aldrich | 311421–50 ML | |
| Software, algorithm | Origin 8 | Company | | |
| Other | ECF-CbrT coordinate file and structure factors | this paper | accession number PDB ID code 6FNP | Crystal structure of ECF-CbrT |

## Molecular methods

For expression, CbrT (LDB_RS00385) was amplified by means of polymerase chain reaction (PCR) using *L. delbrueckii* subsp. *bulgaricus* genomic DNA as a template. For expression of the entire complex, CbrT was inserted into the second multiple cloning site of p2BAD_ECF (*Swier et al., 2016*) with *XbaI* and *XhoI* restriction sites. For expression of solitary CbrT, the gene was inserted with a C-terminal octa-His-tag into pBAD24 (*Guzman et al., 1995*) using *NcoI* and *HindIII* restriction sites. A single glycine (Gly2) was introduced to be in-frame with the start-codon of the *NcoI* restriction site, which is not present in the full complex. All primers used are listed in *Supplementary file 2* and all sequences were checked for correctness by sequencing.

## Expression and membrane vesicle preparation

ECF–CbrT was expressed as described previously (*Swier et al., 2016*) with the following adaptations; plain Luria Miller broth (LB) medium was used and the growth temperature was kept constant at 37°C throughout. After 3 hr of expression, the cells were harvested by centrifugation (20 min, $7,446 \times g$, 4°C) and resuspended in 50 mM KPi, pH 7.5. Cells were either immediately used for membrane vesicles preparation or the resuspended cells were flash frozen in liquid nitrogen and stored at −80°C until use. Membrane vesicles were prepared as previously described (*Swier et al., 2016*)

## ECF-CbrT purification

Crude membrane vesicles containing ECF-CbrT were solubilized in buffer A (50 mM KPi, pH 7.5, 300 mM NaCl, 10% glycerol, 1% (w/v) n-dodecyl-β-D-maltopyranoside (DDM, Anatrace) for 45 min at 4°C under constant movement. Unsolubilized material was removed by centrifugation (35 min, $287,000 \times g$, 4°C), the supernatant was loaded on a BioRad PolyPrep column containing 0.5 mL $Ni^{2+}$-sepharose bed volume (GE healthcare), pre-equilibrated with 20 column volumes (CV) buffer B (50 mM KPi, pH 7.5, 300 mM NaCl, 10% glycerol) and allowed to incubate for 1 hr at 4°C under constant movement. Unbound protein was allowed to flow through and the column was washed with 20 CV of buffer C (50 mM KPi, pH 7.5, 300 mM NaCl, 10% glycerol, 50 mM imidazole, 0.05% DDM). ECF–CbrT was eluted with buffer D (50 mM KPi, pH 7.5, 300 mM NaCl, 10% glycerol, 500 mM imidazole, 0.05% (w/v) DDM) in three fractions of 0.4 ml, 0.75 ml and 0.5 ml, respectively. ECF-CbrT eluted mostly in the second elution fraction that was loaded on a Superdex 200 Increase 10/300 gel filtration column (GE Healthcare) that was equilibrated with buffer E (50 mM Hepes pH 8, 150 mM NaCl, 0.05% DDM). For crystallization, ECF-CbrT was purified following the same protocol but buffers A to D contained 1% DDM. Buffers A to D were supplemented with 0.5 mM hydroxyl-cobalamin (OH-Cbl, Sigma Aldrich), and buffer E contained 10 μM OH-Cbl. For all experiments, the peak fractions were collected, combined and either used directly for reconstitution or concentrated in a Vivaspin disposable ultrafiltration device with a molecular weight cut-off of 30 kDa (Sartorius Stedim Biotech SA) to a final concentration of 6 mg*ml$^{-1}$.

## Construction of the E. coli ΔFEC strain

The *E. coli* strains JW0154 (Δ*btuF::Km^R*), JW3805(Δ*metE::Km^R*) and JW1701(Δ*btuC::Km^R*) from the Keio collection (*Baba et al., 2006*) were purchased from the Coli Genetic Stock Center, Yale. *E. coli* JW0154 (Δ*btuF::Km^R*) was used as the basis for constructing *E. coli*ΔFEC. The kanamycin resistance cassette of JW0154 was removed using the FLP recombinase as described before (*Datsenko and Wanner, 2000*), resulting in *E. coli*ΔF. The *metE::Km^R* locus from JW3805 was introduced in *E. coli*ΔF using P1-mediated generalized transduction as described (*Miller, 1972*; *Thomason et al., 2007*), resulting in *E. coli*ΔFE::*Km^R*. The kanamycin cassette was removed using the FLP recombinase, resulting in *E. coli*ΔFE. The Δ*btuC::Km^R* locus of JW1701 was introduced in *E. coli*ΔFE using P1-mediated generalized transduction, resulting in *E. coli*ΔFEC. Colony PCRs based on three primer pairs (*buF*-locus, 5'-atggctaagtcactgttcagg-3' and 5'-ctaatctacctgtgaaagcgc-3'; *butC*-locus, 5'-atgctgacacttgcccgc-3' and 5'-ctaacgtcctgcttttaacaataacc-3'; *metE*-locus, 5'-atgacaatattgaatcacaccctcg-3' and 5'-ttaccccgacgcaagttc-3') were used to verify Km^R-insertions, the FLP-recombinase-mediated removal of Km^R-markers and the absence of any genomic duplications resulting in the presence of any wild-type *metE*, *btuC* and *btuF* loci.

## Growth assay with *E. coli* ΔFEC strains

The strains carrying various expression vectors were grown overnight at 37°C on LB-agar plates supplemented with 25 µg*ml$^{-1}$ kanamycin and 100 µg*ml$^{-1}$ ampicillin. The composition of the M9-based (47.7 mM Na$_2$HPO$_4$*12H$_2$O, 17.2 mM KH$_2$PO$_4$, 18.7 mM NH$_4$Cl, 8.6 mM NaCl) minimal medium was supplemented with 0.4% glycerol, 2 mM MgSO$_4$, 0.1 mM CaCl$_2$, 100 µg*ml$^{-1}$L-arginine, 25 µg*ml$^{-1}$ kanamycin and 100 µg*ml$^{-1}$ ampicillin. A single colony was picked and used to inoculate a 3 ml to 6 ml liquid pre-culture supplemented with 50 µg*ml$^{-1}$L-methionine (Sigma-Aldrich). The pre-culture was grown ~ 24 hr at 37°C, shaking in tubes with gas-permeable lids (Cellstar). The main cultures were inoculated in a 1:500 inoculation ratio. The main culture had a volume of 200 µl and was supplemented with 0.00001% L-arabinose (Sigma-Aldrich) and either 50 µg*ml$^{-1}$ L-methionine, 1 nM dicyano-cobinamide (Sigma Aldrich), or 1 nM cyano-cobalamin (Acros Organics). The medium was added to a sterile 96 well-plate (Cellstar). The 96-well plate was sealed with a sterile and gas-permeable foil (BreatheEasy, Diversified Biotech). The cultures were grown for 1000 min in a BioTek Power Wave 340 plate reader at 37°C, shaking. The OD$_{600}$ was measured every five minutes at 600 nm. All experiments were conducted as technical triplicates from biological triplicate. To obtain lag-times the averaged growth curves were fitted with the Gompertz-fit in Origin eight and further analyzed as described (*Zwietering et al., 1990*).

## Crystallization and structure determination

Initial crystallization conditions for ECF-CbrT were screened at 4°C using commercial sparse-matrix crystallization screens in a sitting-drop setup and a Mosquito robot (TTP Labtech, UK). Initial crystals were found in the B11 condition (0.2 M KCl, 0.1 M Sodium citrate, pH 5.5, 37% (v/v) Pentaerythritol propoxylate (5/4 PO/OH)) of the MemGold1 HT-96 screen (Molecular Dimensions, UK) that diffracted up to 7.5 Å resolution. Using this condition as a starting point and the detergent (HR2-408) screen (Hampton Research, USA), an optimized condition could be found and contained the detergent ANAPOE®-C$_{12}$E$_{10}$ (Polyoxyethylene(10)dodecyl ether, Hampton Research) as an additive, which yielded crystals diffracting up to 3.4 Å resolution.

X-ray diffraction data were collected from cooled (100 K) single crystals at synchrotron beam lines at the Swiss Light Source (SLS) beamline PXI, Switzerland. The crystals of *apo* ECF–CbrT belong to space group P1 (unit cell parameters: a = 85.47, b = 92.86, c = 105.51, α = 72.568, β = 66.274, γ = 62.893).

To correct for anisotropy, the dataset was treated at the diffraction anisotropy server prior to further processing (*Strong et al., 2006*). Data were processed with XDS (*Kabsch, 2010b*) and scaled with Xscale (*Kabsch, 2010a*). Data collection statistics are summarized in *Supplementary file 1*. The structure of the ECF-CbrT complex was solved by molecular replacement with PHASER MR (*McCoy et al., 2007*) using the *apo* ECF-FolT structure of *L. delbrueckii* (*Swier et al., 2016*) (PDB code 5JSZ) as a search model. For model completion, several cycles of model building with COOT (*Emsley et al., 2010*) and refinement with PHENIX (*Adams et al., 2010*) were performed. The Ramachandran statistics are 72.32% for favoured regions, 26.64% for allowed regions and 1.05% for

outliers. All structural figures in the main text were prepared with an open-source version of pymol (https://sourceforge.net/projects/pymol/).

### Preparation of radiolabeled cobinamide from radiolabeled cobalamin

The required amount of cyano-cobalamin (radiolabeled and unlabeled) was mixed in a 1:1 (v/v) ratio with 70% perchloric acid (Sigma-Aldrich) and incubated for ten minutes at 70°C. To quench the reaction and prevent damage to the substrate, the resulting cobinamide substrate was added to buffer G (as described above), which was additionally supplemented with 5 M NaOH to restore the pH back to 7.5.

### Radiolabeled vitamin B12 transport assay

Purified ECF-CbrT was reconstituted in proteoliposomes as described previously (*Geertsma et al., 2008*). Proteoliposomes were thawed and loaded with 5 mM $MgSO_4$ or $MgCl_2$ and 5 mM $Na_2$-ATP or $Na_2$-ADP through three freeze-thaw cycles. Loaded proteoliposomes were extruded nine times through a polycarbonate filter with a 400 nm pore-size (Avestin), pelleted by centrifugation (267,008 g, 35 min, 4°C) and resuspended in buffer F (50 mM KPi pH 7.5) to 2 $\mu l*mg^{-1}$ lipids. The uptake reaction was started by addition of concentrated and loaded proteoliposomes to buffer G (50 mM KPi pH 7.5, varying concentrations of $^{57}$Co-cyanocobalamin (150 to 300 $\mu Ci*mg^{-1}$, in 0.9% benzylalcohol, MP Biomedicals) in a 1:100 ratio. At elsewhere specified time points 200 $\mu l$ samples were taken transferred into 2 ml ice cold buffer F and filtered over OE67 cellulose acetate filters (GE Healthcare) soaked in Buffer F supplemented with cyanocobalamin (Acros chemicals). The filter was washed with 2 ml ice cold buffer F and transport of radiolabeled substrate was counted in Perkin Elmer Packard Cobra II gamma counter. All uptake assays were performed at 30°C while stirring.

### Substrate-binding assay by isothermal thermal calorimetry (ITC)

ITC measurements were performed using a NanoITC calorimeter (TA Instruments) at 25°C. Membrane vesicles containing CbrT (200 $\mu l$, 10 $mg*ml^{-1}$ in 50 mM KPi, pH 7.5) were added to the Nano-ITC cell. Ligands were prepared in 50 mM KPi, pH 7.5 and titrated into the cell in 1 $\mu l$ injections with 140 s between each injection. Membrane vesicles containing the full-complex ECF-CbrT that does not bind CN-Cbl (10 $mg*ml^{-1}$ in 50 mM Kpi, pH 7.5) were used as a negative control. Data were analyzed with the Nano Analyze Software.

### Data deposition

The atomic coordinates and structure factors have been deposited in the Protein Data Bank, www.pdb.org (PDB ID code 6FNP).

## Acknowledgements

We thank Prof. Dr. AJM Driessen for the use of the setup in the isotope lab and we thank the beamline personnel of PXI at SLS for their technical support. This work was supported by grants from the Netherlands Organisation for Scientific Research (NWO Vici grant 865.11.001 to D-JS and NWO Vidi grant 723.014.002 to AG), the São Paulo Research Foundation (BEPE fellowship 2015/26203-0 to CTP), the European Research Council (ERC starting grant 282083 to D-JS), and the European Molecular Biology Organization (EMBO long-term fellowship ALTF 687–2015 to JAS. and EMBO short-term fellowship ASTF-382–2015 to SR).

## Additional information

### Funding

| Funder | Author |
| --- | --- |
| European Molecular Biology Organization | Joana A Santos Stephan Rempel |
| Nederlandse Organisatie voor Wetenschappelijk Onderzoek | Josy ter Beek Albert Guskov Dirk Slotboom |

| European Research Council | Dirk J Slotboom |
|---|---|

The funders had no role in study design, data collection and interpretation, or the decision to submit the work for publication.

## Author contributions
Joana A Santos, Stephan Rempel, Data curation, Formal analysis, Supervision, Funding acquisition, Acquisition of data, Investigation, Methodology, Writing—original draft, Writing—review and editing; Sandra TM Mous, Investigation, Methodology; Cristiane T Pereira, Investigation, Acquisition of data; Josy ter Beek, Data curation, Investigation, Writing—review and editing; Jan-Willem de Gier, Data curation, Formal analysis, Funding acquisition, Methodology, Writing—review and editing; Albert Guskov, Formal analysis, Supervision, Funding acquisition, Validation, Investigation, Writing—review and editing; Dirk J Slotboom, Conceptualization, Data curation, Formal analysis, Supervision, Funding acquisition, Investigation, Writing—original draft, Project administration, Writing—review and editing

## Author ORCIDs
Joana A Santos  https://orcid.org/0000-0001-8294-3405
Stephan Rempel  http://orcid.org/0000-0003-3569-8229
Albert Guskov  http://orcid.org/0000-0003-2340-2216
Dirk J Slotboom  https://orcid.org/0000-0002-5804-9689

## Decision letter and Author response
Decision letter https://doi.org/10.7554/eLife.35828.016
Author response https://doi.org/10.7554/eLife.35828.017

# Additional files

## Supplementary files
• Supplementary file 1. Data collection, phasing and refinement statistics.
DOI: https://doi.org/10.7554/eLife.35828.010
• Supplementary file 2. Primer list used in this study.
DOI: https://doi.org/10.7554/eLife.35828.011
• Transparent reporting form
DOI: https://doi.org/10.7554/eLife.35828.012

## Data availability
Diffraction data have been deposited in PDB under the accession code 6FNP.

The following dataset was generated:

| Author(s) | Year | Dataset title | Dataset URL | Database, license, and accessibility information |
|---|---|---|---|---|
| Slotboom D | 2018 | Diffraction data from: Functional and structural characterization of an ECF-type ABC transporter for vitamin B12 | http://www.rcsb.org/pdb/search/structid-Search.do?structureId=6FNP | Publicly available at the RCSB Protein Data Bank (accession no: 6FNP) |

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
