## [Decision Letter]

Thank you for submitting your article "Functional and structural characterization of an ECF-type ABC transporter for vitamin B12" for consideration by *eLife*. Your article has been reviewed by two peer reviewers, and the evaluation has been overseen by a Reviewing Editor and Richard Aldrich as the Senior Editor. The following individuals involved in review of your submission have agreed to reveal their identity: Ming Zhou (Reviewer #2); Douglas C Rees (Reviewer #3).

The reviewers have discussed the reviews with one another and the Reviewing Editor has drafted this decision to help you prepare a revised submission.

Summary:

The manuscript by Santos et al. describes a functional and structural characterization of an ECF-type ABC transporter for vitamin B12. ECF-type ABC transporters are a relatively recent addition to the ABC transporter family and consist of two ATPases similar to the ATPases of ABC transporters, with two transmembrane subunits – an ECF-module (energizing/transduction) and an S-component (substrate binding module). An interesting feature is that multiple S-components can interact with a single ECF module. Contributions of this paper include the demonstration that ECF-CbrT functions as a B12 transporter (by complementing an *E. coli* strain lacking the endogenous BtuCDF system (itself an ABC transporter); that ECF-CbrT catalyzes the ATP-dependent transport of B12 derivatives (by transport assays in reconstituted liposomes); that B12 derivatives binding to CbrT with nM affinities; and by determining the crystal structure of ECF-CbrT to 3.4 A resolution in a state likely corresponding to the apo, post-substrate release state. Comparison of this structure to ECF-FolT folate transporter establishes how different S-components interact with the same ECF-module. Together these studies provide a detailed characterization of a newly identified transport system for vitamin B12.

The experimental studies appear to have been carefully conducted, and are generally well documented. The crystal structure and the functional data are of good quality. This study therefore represents an important advance in our understanding of the mechanism of ECF transporters and ABC transporters generally.

Essential revisions:

1) It is not immediately clear how the protein concentration was calibrated in the ITC experiments using the crude membrane vesicles, and how uncertainties in this parameter might influence the thermodynamic data. Please clarify.

2) Please discuss what you propose for the ratio of ATP hydrolyzed to ligand transported, and the basis for that proposal. If feasible, can this stoichiometry be determined from experiments such as those depicted in Figure 3?

---

## [Author Response]

Essential revisions:1) It is not immediately clear how the protein concentration was calibrated in the ITC experiments using the crude membrane vesicles, and how uncertainties in this parameter might influence the thermodynamic data. Please clarify.

We now clarify these points in the text:

“Although the use of crude membrane vesicles precluded the determination of the number of binding sites (the concentration of CbrT in the membrane vesicles is unknown), the thermodynamic values (*Kd, ΔH* and *ΔS*) derived from the ITC measurements do not depend on this number. Assuming that CbrT has a single substrate binding site (consistent with the spectral properties, Figure 4A), the expression level of CbrT in the membranes could be calculated and we found that CbrT accounted for ~0.9% (w/w) of the protein content in the membrane.”

2) Please discuss what you propose for the ratio of ATP hydrolyzed to ligand transported, and the basis for that proposal. If feasible, can this stoichiometry be determined from experiments such as those depicted in Figure 3?

We added the following text:

“While this experiment shows that transport is strictly ATP- dependent, similar to what was found for other ECF transporters (Swier et al., 2016; ter Beek et al., 2011a), the ratio between ATP molecules hydrolysed and Cbl molecules transported cannot be derived from this data. […] Nonetheless, the EcfA and EcfA’ subunits contain all the motifs to form functional ATPases, and therefore we speculate that transport of Cbl is coupled to the hydrolysis of two ATP molecules.”